# Discovering Weight Initializers with Meta Learning

**Dmitry Baranchuk**  DMITRY.BARANCHUK@GRAPHICS.CS.MSU.RU
*Yandex, Russia*
*National Research University Higher School of Economics, Russia*

**Artem Babenko**  ARTEM.BABENKO@PHYSTECH.EDU
*Yandex, Russia*
*National Research University Higher School of Economics, Russia*

## Abstract

Deep neural network training largely depends on the choice of initial weight distribution. However, this choice can often be nontrivial. Existing theoretical results for this problem mostly cover simple architectures, e.g., feedforward networks with ReLU activations. The architectures used for practical problems are more complex and often incorporate many overlapping modules, making them challenging for theoretical analysis. Therefore, practitioners have to use heuristic initializers with questionable optimality and stability. In this study, we propose a task-agnostic approach that discovers initializers for specific network architectures and optimizers by learning the initial weight distributions directly through the use of Meta-Learning. In several supervised and unsupervised learning scenarios, we show the advantage of our initializers in terms of both faster convergence and higher model performance. The PyTorch implementation of our algorithm is available online[1].

## 1. Introduction

A key ingredient to the success of deep learning is its flexibility. Depending on the task at hand, a deep learning practitioner can use convolutional, recurrent, attentive, graph-based, or hundreds of other layer types. The layers can be combined in a myriad of ways ranging from stacking layers sequentially to using residual or densely connected architectures (He et al., 2016; Huang et al., 2017), to highly specialized architectures with multiple pathways, each processing a subset of input features (Cheng et al., 2016; Vaswani et al., 2017).

However, designing a deep learning model is not limited to the architecture. One also needs to choose a training protocol and decide on the initialization. This choice can significantly affect training speed, stability (Mishkin and Matas, 2015), and, for some architectures, the ability to train at all (Xiao et al., 2018; Zhang et al., 2019a; Xu et al., 2019).

The majority of real-world architectures rely on standard heuristic initializers (Glorot and Bengio, 2010; He et al., 2015). While these heuristics fit most training scenarios, they can often be surpassed by the initializers designed for the specific architecture in use (Xiao et al., 2018; Zhang et al., 2019a; Xu et al., 2019; Zhang et al., 2019b; Le et al., 2015). The more comprehensive discussion of the related work is deferred to Appendix A.

Discovering effective task-specific initializers is an arduous task that requires human intuition and theoretical insight. This task becomes more and more complex as we invent more sophisticated network architectures. In this work, we argue that this task can and should be automated to save valuable human time.

---

1. `https://github.com/yandex-research/learnable-init`

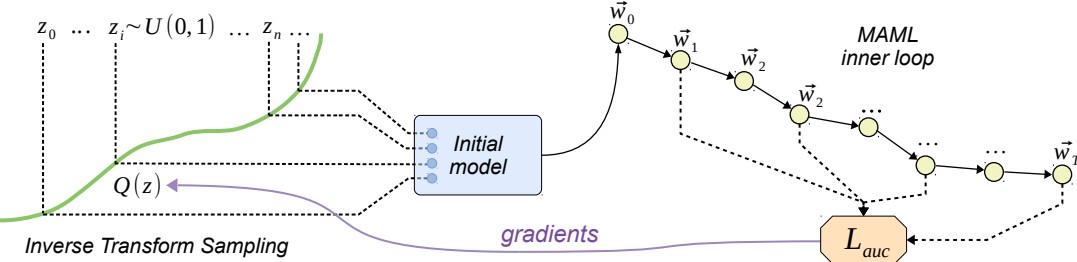

Figure 1: **DIMAML-PLIF training scheme:** layer weights $\theta_i$ are obtained by sampling $z \sim U(0,1)$ and computing $Q_{s,b}(z)$, for each layer in the network. Then, during the inner training process, DIMAML computes the loss function on validation data and backpropagates the loss w.r.t. the parameters $s$, $b$ of weight distribution.

We propose a method for **D**iscovering **I**nitializers with **M**odel **A**gnostic **M**eta-**L**earning (DIMAML). Instead of optimizing model weights, DIMAML learns the probability distribution from which these weights are generated. The goal is to learn initializers that optimize the training performance for the chosen architecture without being task-specific.

In other words, DIMAML learns initialization strategies that capture the specifics of the model architecture and the training procedure but can be applied outside the specific task that was used for meta-learning. Therefore, it is possible to learn the initializers once and distribute them alongside the model architecture. We summarize the contributions of our paper as follows:

1. We propose DIMAML — an automated approach that learns initial parameter *distributions* for a given architecture by directly optimizing its training performance;

2. We evaluate our approach on several architectures for image and sequence processing tasks and demonstrate that the initialization strategies learned by DIMAML match or outperform existing alternatives;

3. We examine universality of the learned initializers by measuring their ability to generalize to unseen tasks, including different data distributions and training protocols.

## 2. Method

The main idea of DIMAML is to train weight initializers by backpropagating through the training procedure. When learning initializers for a given neural network, we assume initial weights $\theta_{init}$ to be an i.i.d. sample from $p_\psi(\theta_{init})$ distribution with trainable parameters $\psi$.

For each DIMAML update, we sample $\theta_{init} \sim p_\psi(\theta_{init})$, then train these weights for some task-specific objective $\mathcal{L}$ using gradient descent. Finally, we compute $\partial \mathcal{L}(\theta_{trained})/\partial \psi$ by backpropagation and update the meta-parameters $\psi$. This procedure requires that $p_\psi(\cdot)$ is reparameterizable (Kingma and Welling, 2014), i.e. sampling $\theta_{init} \sim p_\psi(\theta_{init})$ is differentiable w.r.t. $\psi$. We study two such distributions:

**Normal initializers.** one popular choice of initializer is a Normal distribution with parameters $\psi = \{\mu, \sigma\}$. Under such parameterization, initial weights for layer $l$ are sampled as: $\theta_l = \mu_l + \sigma_l \cdot z$, where $z \sim N(0,1)$. Conveniently for us, this sampling scheme is already differentiable w.r.t. $\mu_l, \sigma_l$.

**PLIF initializers.** However, Normal initialization limits the space of distributions that DIMAML can learn. Thus, we propose to learn the shape of weight distributions in the form

of their quantile functions[2]. By definition, a quantile function $Q$ is non-decreasing function with support $(0, 1)$. A natural way to model this kind of functions is **PLIF** (Ganea et al., 2019) — a class of parametric **p**iecewise **l**inear **i**ncreasing **f**unctions that was developed to improve the expressiveness of softmax layers in natural language processing tasks.

In order to parameterize quantile functions, we define **PLIF** on the $[0, 1]$ range with $K+1$ equally wide regions: $m_i = \frac{i}{K}, \forall 0 \leq i \leq K$. This function is piecewise linear, meaning that $f(x) = s_i \cdot x + b_i, \forall x \in [m_i, m_{i+1})$, where $s_i > 0$ is the slope of the linear function on the interval $[m_i, m_{i+1})$. We chose this parameterization because it can approximate continuous quantile functions arbitrarily well with enough slope regions.

In order to sample weights $\theta_{init}$ with distribution function $F$, we sample uniform random variable $z \sim U(0, 1)$, then compute $\theta_{init} = F^{-1}(z) = Q_{s,b}(z)$. Being computationally efficient, this procedure is also differentiable w.r.t. trainable parameters $\psi = \{s, b\}$. Similar to before, we learn several independent quantile functions, one per each weight tensor.

### 2.1 Training algorithm

The most natural way to evaluate a weight initializer is to train a neural network with that initializer. Depending on the particular machine learning setup, we can choose initializers based on their training speed, stability, or the final performance of the trained model.

Following this intuition, DIMAML explicitly maximizes the training performance of neural networks as a function of their initializers. Formally, consider a neural network $f_{\theta_1,\theta_2,\ldots,\theta_m}(x)$ with $m$ trainable weight tensors. We denote the distribution for initializing $i$-th weight tensor as $p_\psi^i(\theta_i)$.

First, we generate initial weight matrices $\theta_i \sim p_\psi^i(\theta_i), \forall i \in (1, \ldots, m)$. Then, we apply the gradient descent algorithm (e.g., SGD or Adam (Kingma and Ba, 2014)) for a certain number of iterations to emulate the training process and record model performance on validation samples.

The final DIMAML objective is the average validation loss measured on intermediate training steps: $\mathcal{L}_{auc} = \frac{1}{N} \sum_{i=1}^N \mathcal{L}_{step_i}(x_{val}, y_{val})$. Intuitively, minimizing this objective is equivalent to minimizing the area under the validation loss curve, which should lead to faster training and better performance of the final model (Antoniou et al., 2019).

We exploit the ideas from Model-Agnostic Meta-Learning (Finn et al., 2017) to compute the gradients of $\mathcal{L}_{auc}$ w.r.t. weight initializers $p_\psi(\cdot)$ by backpropagating through the training loop. These gradients are used to train the distribution parameters $\psi$ by a meta-optimizer, which is also a gradient descent algorithm.

Since we want DIMAML to be task-agnostic, we measure its transferability to unseen tasks. In each experiment, we use a single $\mathcal{T}_{train}$ task to learn initializers, then evaluate on different $\mathcal{T}_{test}$ tasks. Note that DIMAML can deal with an arbitrary number of $\mathcal{T}_{train}$ tasks.

**Memory Efficient MAML**. One of the limitations of MAML is large memory footprint when applied to commonly used neural network architectures and datasets. In DIMAML, we mitigate this issue through the use of gradient checkpointing (Griewank and Walther, 2000; Chen et al., 2016). This technique stores only 1 in $m$ optimizer states in device memory, recomputing intermediate steps on the fly. Therefore, one can fit more optimizer steps in the same GPU memory. Gradient Checkpointing fits well with MAML because its computation

---

2. or equivalently, inverse CDF functions

| $\mathcal{T}_{test}$ | CelebA | | | Tiny ImageNet | | |
|---|---|---|---|---|---|---|
| Epoch | 10 | 50 | 100* | 10 | 50 | 100* |
| Kaiming | 0.138±.001 | 0.122±.001 | 0.117±.001 | 0.281±.001 | 0.261±.003 | 0.257±.001 |
| DeltaOrthogonal | 0.143±.001 | 0.126±.001 | 0.120±.001 | 0.288± .001 | 0.264±.001 | 0.259±.001 |
| MetaInit | 0.175±.007 | 0.135±.002 | 0.128±.002 | 0.331±.023 | 0.279±.002 | 0.273±.002 |
| DIMAML-Normal | **0.120±.001** | **0.112±.000** | **0.109±.001** | **0.259±.001** | **0.252±.001** | **0.250±.001** |
| DIMAML-PLIF | **0.121±.001** | **0.112±.000** | **0.110±.000** | **0.260±.001** | **0.252±.001** | **0.250±.000** |
| $\mathcal{T}_{test}$ | AnimeFaces | | | AnimeFaces Shuffled Pixels | | |
| Epoch | 10 | 50 | 100* | 10 | 50 | 100* |
| Kaiming | 0.471±.002 | 0.383±.001 | 0.369±.001 | 0.823±.005 | 0.643±.005 | 0.590±.002 |
| DeltaOrthogonal | 0.496±.005 | 0.392±.001 | 0.377±.001 | 0.842±.005 | 0.743±.016 | 0.652±.012 |
| MetaInit | 0.552±.013 | 0.437±.021 | 0.398±.007 | 0.859±.005 | 0.792±.026 | 0.708±.031 |
| DIMAML-Normal | **0.386±.003** | **0.352±.003** | **0.344±.001** | **0.814±.013** | **0.570±.003** | **0.546±.002** |
| DIMAML-PLIF | **0.386±.006** | **0.350±.001** | **0.344±.001** | **0.812±.008** | **0.567±.005** | **0.545±.003** |

Table 2: Comparison of autoencoder models with different initializers in terms of mean squared error. Evaluation is performed on $\mathcal{T}_{test}$ datasets according to Table 1. DIMAML significantly outperforms baselines even on AnimeFaces with shuffled pixels which is very different from $\mathcal{T}_{train}$. (*) corresponds to convergence.

graph has natural "choke points" after each optimizer step. In our experiments, we place gradient checkpointing every $5 - 10$ steps of the inner optimizer, allowing us to perform ($\sim 10 - 100\times$) more steps within the same GPU memory budget.

## 3. Experiments

To thoroughly investigate the performance of DIMAML, we apply it to several commonly used neural network architectures: convolutional autoencoders, residual networks (He et al., 2016) and language models based on LSTM (Hochreiter and Schmidhuber, 1997). For each architecture, we compare the weight distributions learned by DIMAML with popular initialization techniques as well as another Meta Learning approach MetaInit (Dauphin and Schoenholz, 2019).

### 3.1 Autoencoders

Our first setup considers training a convolutional autoencoder which trains reasonably well with standard initializers like Kaiming. In this experiment we start with a default training protocol and examine whether DI-MAML is able to discover a better initializer.

| $\mathcal{T}_{train}$ | $\mathcal{T}_{test}$ |
|---|---|
| CelebA 64x64 | Tiny Imagenet |
| Tiny Imagenet | CelebA 64x64 |
| Tiny Imagenet | AnimeFaces |
| Tiny Imagenet | AnimeFaces Shuffled Pixels |

Table 1: $\mathcal{T}_{train}$ and $\mathcal{T}_{test}$ dataset pairs used for autoencoder evaluation.

The autoencoder model follows the architecture proposed in (Ghosh et al., 2020). The experiment is performed on two standard image datasets: Tiny Imagenet (Le and Yang, 2015) and CelebA (Liu et al., 2015) resized to 64x64 resolution. We also consider the AnimeFaces dataset[3] for evaluation because it differs significantly from natural images. We also evaluate on AnimeFaces with randomly permuted pixels to further test DIMAML's task-agnosticism. The pairs of $\mathcal{T}_{train}$ and $\mathcal{T}_{test}$ for autoencoder architecture are presented in Table 1. The training and meta-training protocols are described in Appendix B.1.

---

3. https://www.kaggle.com/soumikrakshit/anime-faces

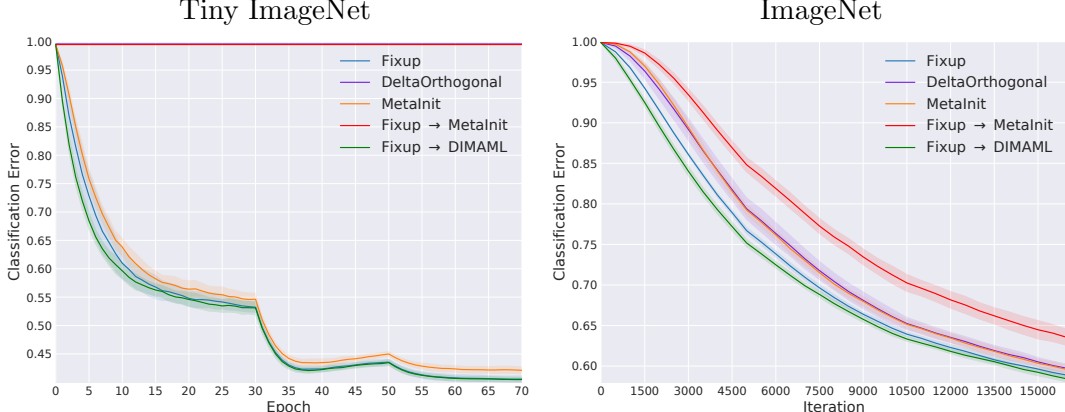

Figure 2: Classification performance evaluation on Tiny ImageNet (Left) and ImageNet (Right) for ResNet-18. During the first 25 epochs on Tiny ImageNet and 3 epochs on ImageNet, DIMAML is superior over all baselines and then converges similar to top-performing Fixup reaching the same optima.

After meta-learning converges, we evaluate the obtained initialization by inspecting the model performance over longer training sessions of 10, 50, and 100 epochs. As main baselines, we evaluate Kaiming (He et al., 2015) and MetaInit (Dauphin and Schoenholz, 2019) initializations. Delta Orthogonal (Xiao et al., 2018) is also considered, although it was designed for deeper convolutional networks without batch normalization. The meta-optimization starts from the Kaiming initializer.

The evaluation results are presented in Table 2. From the quantitative standpoint, initializer distributions learned by DIMAML demonstrate significantly faster and tighter convergence. DIMAML also outperforms baseline on AnimeFaces and AnimeFaces with shuffled pixels, both of which are significantly different from $\mathcal{T}_{train}$. We also note that DIMAML with PLIF demonstrates similar performance as DIMAML-Normal. In Appendix C.1, we aim to explain the advantage of the DIMAML initializers for the autoencoder model.

## 3.2 Residual Networks

In order to evaluate how DIMAML handles more complex training scenarios, we apply it to ResNet-18 on top of the Fixup initialization (Zhang et al., 2019b). Fixup is a ResNet-specific initialization that allows for faster and more stable training without the use of normalization layers. In Fixup, half of the weight tensors are initialized with zeros, while the other half uses scaled Kaiming initializers. In this experiment, we apply DIMAML to adjust non-zero initializers and discover whether there is a potential for further improvement.

DIMAML initializers are trained on the CIFAR100 dataset and evaluated on TinyImageNet and ImageNet. Note that CIFAR100 has much smaller image resolution and number of classes compared to both $\mathcal{T}_{test}$. For CIFAR100 and Tiny Imagenet datasets, we adapt ResNet-18 to deal with low resolutions by replacing the first convolutional and maxpooling layers with a single 3×3 convolution. For ImageNet evaluation, we use the original model.

We perform a comparison of the learned initializers with Fixup, DeltaOrthogonal and MetaInit methods. MetaInit is applied on top of Fixup and Kaiming initializations as proposed in (Dauphin and Schoenholz, 2019). We present test classification error curves in

| $\mathcal{T}_{test}$ | Wikitext2 | | | PennTreebank | | |
|---|---|---|---|---|---|---|
| Epoch | 10 | 50 | 100* | 10 | 50 | 100* |
| Kaiming | 1.983±.009 | 1.843±.006 | 1.801±.002 | 1.492±.004 | 1.377±.002 | 1.352±.001 |
| Orthogonal | 2.017±.007 | 1.849±.003 | 1.806±.004 | 1.528±.009 | 1.383±.003 | 1.354±.002 |
| MetaInit | 1.907±.005 | 1.819±.002 | 1.792±.003 | 1.477±.003 | 1.382±.004 | 1.359±.005 |
| DIMAML-Normal | **1.879±.005** | **1.812±.003** | **1.782±.003** | **1.454±.003** | **1.372±.002** | **1.345±.003** |
| DIMAML-PLIF | **1.876±.003** | **1.810±.004** | **1.784±.004** | **1.452±.003** | **1.369±.001** | **1.344±.002** |

Table 3: Performance of the character-level language model in bits-per-character (bpc). DIMAML initial distributions speedup the training and converges to better optima for the same number of epochs. (*) corresponds to convergence.

Figure 2. For this experiment we omit DIMAML-PLIF since it matches the performance of DIMAML-Normal almost exactly as in Section 3.1.

**TinyImageNet, Left**. We observe that DIMAML demonstrates faster training during the first 25 epochs and then coincides with Fixup baseline until convergence. In turn, MetaInit based on Kaiming fares significantly worse than Fixup and DIMAML on the entire curve, while MetaInit based on Fixup does not converge at all.

**ImageNet, Right**. To test the transferability of the learned initializers, we run the standard training protocol for Fixup on ImageNet. This task is different from $\mathcal{T}_{train}$ in both data and training protocol. The error curves are visualized over 3 full epochs. Again, DIMAML outperforms the baselines in the early stages and matches Fixup in convergence. Note that DIMAML demonstrates similar behavior as on TinyImageNet despite larger image resolution, different training protocol and modified architecture.

### 3.3 Language Modeling

In this final experiment, we apply DIMAML to discover initial distributions for recurrent neural networks. To do this, we formulate a character-level language modeling problem. Both training and evaluation are performed on common benchmark datasets for language modeling: Wikitext2 (Merity et al., 2016) and PennTreebank (Marcus et al., 1993). DI-MAML initializers trained on Wikitext2 are evaluated on PennTreebank and vice versa. The model architecture and inner loop optimization setting are described in Appendix B.3.

As for baselines, we tried Kaiming initializers, but for the embedding layer, which is initialized as $\mathcal{N}(0, 1)$; we also consider MetaInit and orthogonal initialization for weight tensors, which deal with previous hidden states as proposed in (Henaff et al., 2016).

Overall, the training performance of DIMAML initializers, see Table 3, is superior to all baselines both in terms of faster training and the final quality of the converged model. We also discuss the learned quantile functions of DIMAML distributions in Appendix C.2.

### 4. Conclusion

In this paper, we addressed the learning of the optimal weight initializer for a given objective, model architecture and optimization method. Unlike existing works, DIMAML explicitly learns initializers that maximize both convergence rate and final model performance. DI-MAML can be used for any DNN architecture and is much more universal than existing heuristic-based initializers. We empirically confirm the advantage of learnable initializations on several standard learning scenarios for different data domains. Further discussions on DIMAML performance, primary use cases and limitations are provided in Appendix E.

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

# Appendices

## Appendix A. Related work

### A.1 Weight initialization

Choosing initial weights is a long-standing problem in deep learning. Since the number of works on weight initialization is enormous, we only briefly describe the main ideas underlying this field.

**Traditional initializers.** Earlier neural networks were typically initialized with small random numbers from normal or uniform distribution (LeCun et al., 1989). However, for deeper networks, this initialization has proven inefficient. A poorly chosen initializer can cause network activations to explode or vanish (He et al., 2016; Hanin and Rolnick, 2018). Since then, the community put much effort into the design of initial distributions that ensure the stability of activations (Glorot and Bengio, 2010; He et al., 2015). These initializers are often the default choice in existing deep learning packages (ten, 2015; Paszke et al., 2019). However, they were originally developed for a narrow subset of neural networks. For instance, some works require a certain number of layers (Chen and Bastani, 1990) or loss function (Ghods et al., 2020). The effectiveness of such initializers in complex real-world architectures is not guaranteed.

**Model-specific initializers.** Several works have addressed initialization for specific architectures, e.g. convolutional (Xiao et al., 2018), residual (Zhang et al., 2019b; Arpit et al., 2019; Yang and Schoenholz, 2017), popular recurrent models (Pascanu et al., 2013; Aghajanyan, 2017; Chen et al., 2018; Gilboa et al., 2019), and others (Zhang et al., 2019a; Xu et al., 2019; Blumenfeld et al., 2019; Yang et al., 2019). Alas, coming up with these initializations requires careful analysis of the architecture and the data distribution. When deep learning community develops a novel model architecture, it may take months or years for effective initialization strategies to emerge.

**Data-aware initializers.** Another line of research focuses on data-aware initialization (Mishkin and Matas, 2015; Krähenbühl et al., 2015). These approaches dynamically adjust initial weights on a sample of inputs to have the desired statistical properties. In general, data-aware initialization is more flexible to the choice of layer structure and activation functions, but it can still be impractical for more complex models such as recurrent neural networks or architectures with weight sharing. Moreover, data-aware initializations are essentially heuristics: they focus on attaining certain statistical properties instead of directly optimizing the convergence rate or the performance of the initialized model.

### A.2 Meta-learning

Meta-learning is a family of methods that aims to learn algorithms for training machine learning models. These methods were shown to be successful in a large number of problems, such as few-shot learning (Finn et al., 2017; Nichol and Schulman, 2018), learnable optimizers (Andrychowicz et al., 2016), and reinforcement learning (Houthooft et al., 2018).

**Model-Agnostic Meta-Learning (MAML)** (Finn et al., 2017) is a popular technique that learns neural network weights by explicitly optimizing their ability to be trained by gradient descent. In a nutshell, MAML is based on the idea that a gradient-based neural

network training is itself a differentiable function that can be backpropagated through. Using this observation, one can explicitly learn initial weights that are most appropriate for training with a specific learning algorithm, such as SGD or Adam (Kingma and Ba, 2014).

This idea inspired a lot of subsequent research that tackle its practical issues. A recent study (Antoniou et al., 2019) presents several recipes that stabilize training and improve generalization performance. Other works (Nichol et al., 2018a; Nichol and Schulman, 2018) develop light-weight MAML approximations that use only first-order information. Finally, (Rajeswaran et al., 2019) proposes a more efficient way to obtain meta-gradients via implicit differentiation.

### A.3 Meta-learning for initialization

To the best of our knowledge, only one recent work addresses model initialization from the meta-learning perspective - MetaInit (Dauphin and Schoenholz, 2019). The authors propose to train initial weight norms to minimize *gradient quotient*: the change in gradient direction between initial iterations of SGD. This allows MetaInit to recover from pathological initializations and, in some cases, improve model performance. However, minimizing the gradient quotient does not guarantee optimal training performance, and we argue that there is still room for improvement. We provide a more detailed comparison between DIMAML and MetaInit in Section 3.

## Appendix B. Experimental details

### B.1 Autoencoders

For all datasets, we use 64-dimensional latent space. The autoencoder is trained to minimize mean squared error using SGD with momentum of 0.9, learning rate of $10^{-2}$, and mini-batch sizes equal to 128. The number of steps in the inner loop is 600.

Weight distributions are meta-optimized using Adam with learning rate of $10^{-4}$ and $\beta_1=0.9$ and $\beta_2=0.997$. Note that while our model contains batch normalization layers (Ioffe and Szegedy, 2015), the statistics of these layers are not meta-learned, but are instead reset at the start of every training run.

### B.2 Residual Networks

DIMAML emulates the following training protocol: the optimizer is SGD with Nesterov momentum of 0.9, learning rate of 0.1 and weight decay of $5e-4$. The inner loop runs for 200 optimization steps with a batch size of 128. The meta-optimizer is Adam with learning rate of $10^{-4}$ and $\beta_1=0.9$ and $\beta_2=0.997$.

### B.3 Language Modeling

As a character-level language model, we consider an architecture consisting of an embedding layer, two-layer LSTM module, and one fully connected layer to compute logits. The hidden size of the LSTM module is 256, the dimensionality of embeddings is 128. The inner loop runs for 200 Adam optimization steps with learning rate of $10^{-3}$ and batch size of 128. The meta-optimizer is Adam with learning rate of $3\times10^{-4}$ and $\beta_1=0.9$ and $\beta_2=0.997$.

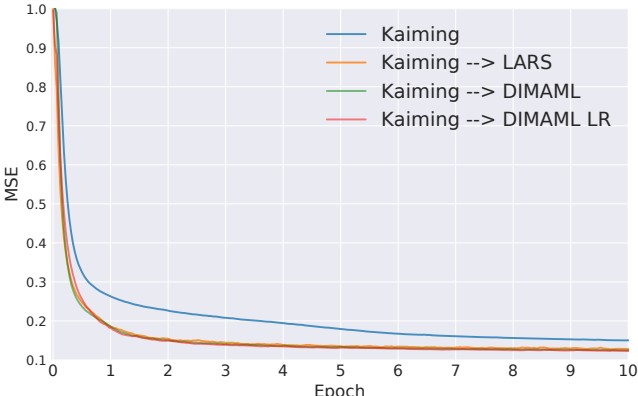

Figure 3: Autoencoder MSE curves for Kaiming and DIMAML initializers during the training on CelebA. **DIMAML LR** transfers individual layer update rates obtained during DIMAML's training to the baseline initialization. While DIMAML converges much better than the baseline, this appears to be solely due to the layer-wise update rates for autoencoder model.

## Appendix C. Analysis

### C.1 Autoencoders

To explain the advantage of the learned initializers for the autoencoder model, we first examine the quantile functions of distributions learned on Tiny ImageNet dataset and present them in Figure 5. After a coarse examination of these distributions, we observe that the learned distributions have only slight changes in shape but significantly lower variances compared to the Kaiming initializer.

Since the autoencoder architecture uses Batch Normalization, model activations are invariant to the scale of weights. However, the rate at which these weights are updated (Santurkar et al., 2018; Bjorck et al., 2018) is not invariant. Decreasing the magnitude of initial weights will accelerate training for that particular layer. To illustrate this phenomena, we report the individual update rates for every weight tensor. We estimate these quantities as a ratio between gradient and weight norms after each training step on the CelebA dataset (see Figure 4). We observe that DIMAML initialization causes most layers to train faster especially in the early stages of training.

Finally, we test whether DIMAML initialization acquired any side-effects that were not covered by the above analysis. We isolate the effect of individual update rates by training with Kaiming initializer and artificially rescaling gradients to match DIMAML update rates from Figure 4. The training curves on the CelebA dataset are presented in Figure 3. We observe that transferring DIMAML individual update rates at each training step demonstrates almost the same performance as the original DIMAML. Therefore, we conclude that for this simple model, DIMAML succeeds mostly due to the individual learning rates, which also explains its task-agnostic properties (Table 2).

The layer-wise learning rates are reminiscent to those produced by LARS (You et al., 2017) optimizer. After tuning the "trust" coefficient, we observe the similar performance compared to DIMAML. Interestingly, the learned initialization produces the learning rate schedule achieved by LARS, which manually tunes learning rates at each training iteration.

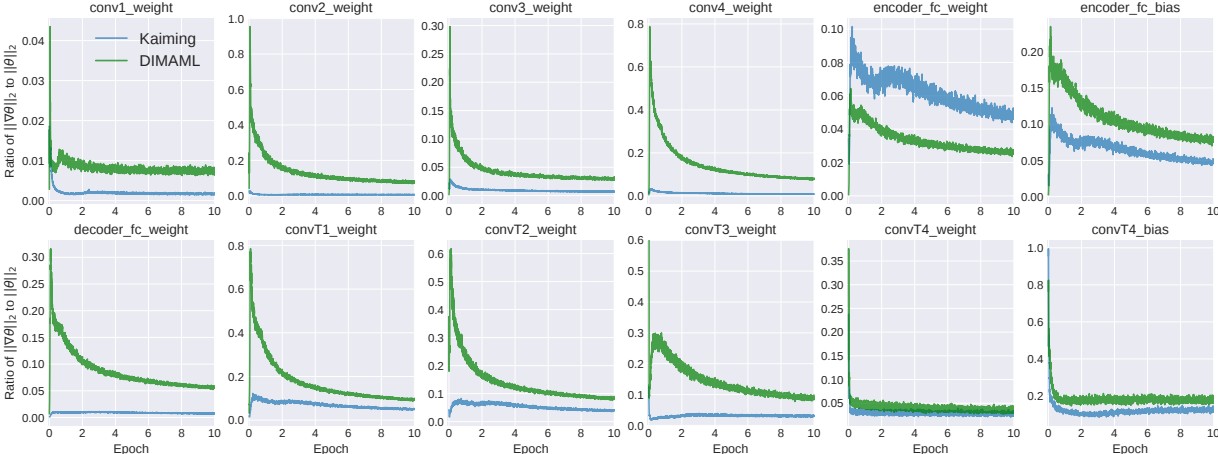

Figure 4: Plots of adjusted individual learning rates for Kaiming and DIMAML initializers for autoencoder architecture trained on Tiny ImageNet and evaluated on CelebA. DIMAML learns initializers that change the relative pace at which different layers are trained.

## C.2 Language Modeling

We also report learned quantile functions of DIMAML distributions in Figure 6. These distributions differ from Kaiming not only in terms of mean and variance, but also shapes, e.g. "logits" layer transformed from uniform to the "sigmoid"-like quantile function. However, when we replaced the learned shapes with corresponding normal distributions difference in performance was negligible.

## Appendix D. Alternative meta-learning techniques.

There are first-order MAML approximations like FOMAML (Nichol et al., 2018b) and Reptile (Nichol and Schulman, 2018). These methods do not take into account second derivatives, resulting in much faster meta-training iterations. However, in our experiments, both methods converged to significantly weaker initializations compared to second order MAML. We attribute this to the coarse gradient approximations used in both (Nichol et al., 2018b; Nichol and Schulman, 2018) that are insufficient for our task.

On the other hand, implicit MAML (Rajeswaran et al., 2019), computes exact meta-gradients through implicit differentiation. However, the technique requires training $f(\cdot)$ to convergence on each meta-learning step, which is infeasible for full-size neural networks.

## Appendix E. Discussion

- **PLIF vs Normal.** In our experiments, we observe, that DIMAML-PLIF does not utilize the given degree of freedom and only slightly changes the shapes of distributions. These changes affect neither the training speed nor the total performance. This observation is consistent with common knowledge in the community that the shapes of initial distributions do not make a difference if the weights in each layer are i.i.d. and have light-tailed distributions.

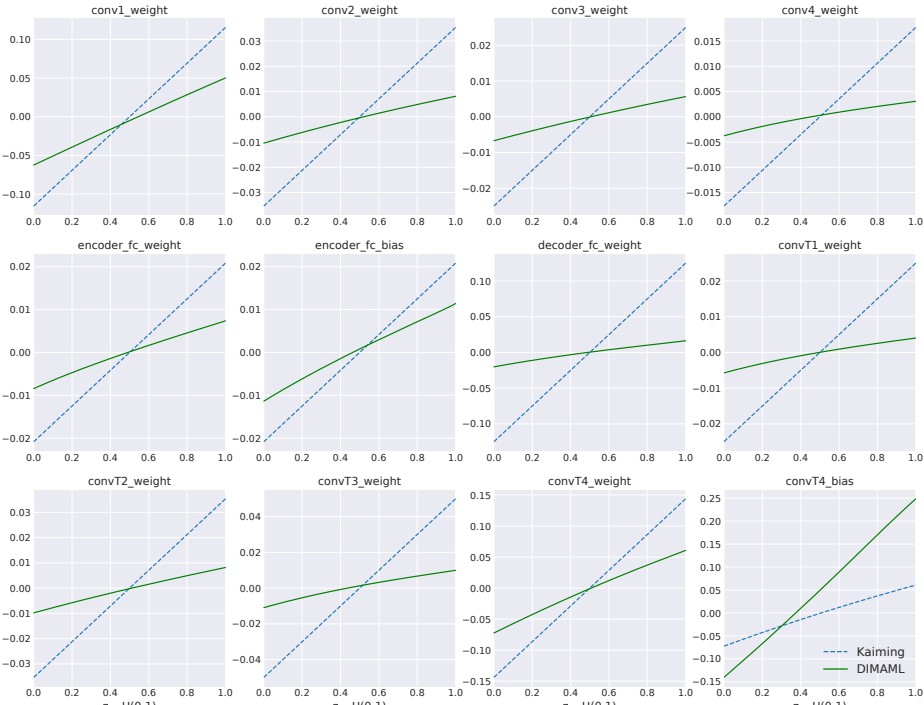

Figure 5: The distributions for Kaiming uniform and DIMAML initializers for autoencoder model presented as quantile functions trained on CelebA. The learned distributions have much lower variance leading to better convergence. The shapes of the learned distributions do not affect the performance.

- **Recovering from bad initializations**. If the base initialization does not cause explosive gradients then DIMAML is able to recover from bad initialization. However, there are no formal guarantees for that. Therefore, when one needs to find `some` working initialization we recommend using MetaInit. Note that MetaInit can be used in conjunction with DIMAML: the former recovers from the bad initialization, and the latter optimizes the MetaInit output to improve training performance even further.

- **Primary use cases**. Firstly, one can use DIMAML for the training protocols that are not properly tuned. DIMAML will work around its issues by speeding up the training and reaching better optima, as we demonstrated in Section 3.1 and Section 3.3. Otherwise, DIMAML is likely to speed up some training regions leading to faster convergence. Moreover, DIMAML can be theoretically valuable by providing some insights for novel tasks and neural network architectures.

- **Scalability.** In principle, DIMAML can work with arbitrary models. However, due to memory and computational limits, it might be unable to emulate enough training steps to dispose of the short-horizon bias (Wu et al., 2018) and improve the total convergence. On the other hand, further increasing the number of steps can lead to unstable meta-training.

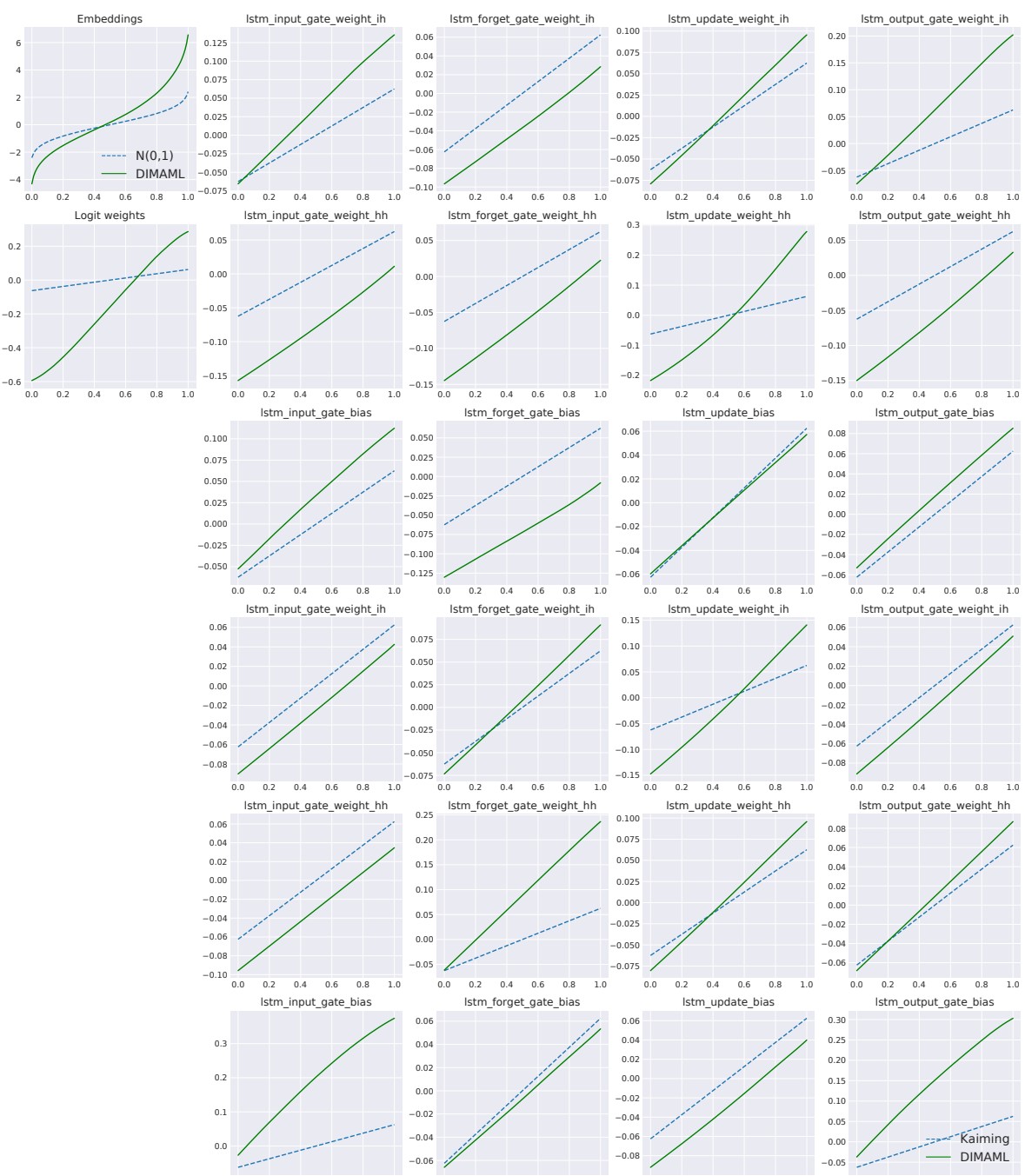

Figure 6: The quantile functions of the baseline ($\mathcal{N}(0,1)$ distribution for the Embedding layer and Kaiming uniform for others) and learned initial distributions for the character-level language model. The initializers are learned on the PennTree-bank dataset. 'ih' and 'hh' correspond to weight tensors that process inputs and previous hidden states, respectively.

