# OpenReview forum: "Discovering Weight Initializers with Meta Learning"
_ICML.cc/2021/Workshop/AutoML — AutoML@ICML2021 Oral_

### Official Review · Reviewer_qVux · 2021-06-11
**Interesting idea, nicely fitting the workshop scope**

**Rating:** 8
**Confidence:** 5

**Review:**

The authors propose to employ a MAML-like method for leaning a probability distribution which is used to initialize neural network weights.  The initial distribution is parametrized as either a normal distribution or a more powerful class of distributions, namely parametric linear increasing functions. The reparameterization trick is used to make both classes differentiable w.r.t. their parameterization. Optimal parameters of the distribtions are then meta-learned with MAML. As a meta-learning objective function, the authors use the AUC of the validation loss.

Experiments are conducted for training Auto-Encoders,  ResNets on Cifar-100 as well as language modelling. The proposed method consistently outperforms baselines, however typically by a small margin.

Overall, an interesting and reasonable approach in a field with little other work, with thorough empirical evaluation. The paper nicely fits the scope of this workshop.

Some comments/questions:
- It is not so clear to me for how many iterations networks are trained during meta training since MAML typically is problematic when training for a typically amount of iterations in the order of thousands or millions. The authors use a trick to reduce memory requirements, but memory and computational stability should still be problematic.
- Did the authors consider using alternative methods that scale less poorly with the number of training iterations, such as REPTILE? Alternatively, gradient estimators such as REINFORCE could be evaluated.
- It would be interesting to see how the learning curves change for different meta-learning objective functions (e.g. aiming at either optimizing the final performance or aiming at achieving very strong performance very early )

---

### Official Review · Reviewer_71wN · 2021-06-15
**learning network initializer with MAML: an intuitive method with exsentive experimental support; concern about time cost**

**Rating:** 8
**Confidence:** 4

**Review:**

This paper proposes a MAML-based meta learning method to learn the optimal weight initializer for a specific model architecture and optimizer in a task-agnostic way, so that the learned initializer can generalize to new datasets while using the same model architecture. The method is clear and intuitive, with carefully designed parameterization of the initializer to make gradient-based meta learning feasible.

The authors conduct extensive experiments in different domains to validate the advantage of their method compared to other heuristic initializers and meta-learning based initializer w.r.t. convergence speed and performance. They have also carefully analyzed why their method gives better results compared to the baselines in the appendix.

Generally, I think this is a good paper and should be accepted. My main concern is about the computational cost. As this method currently needs to be retrained from scratch for each new model architecture (generalization across architectures is considered in section 3.2, but in a very limited way), it is important that the learned initializer can generalize to many different test tasks and the speedup on these tasks are significant enough to compensate for the additional computational cost introduced by learning the initializer on the training task. Thus it would be more convincing to me if the computational cost of the training and test cost are also reported, and more test datasets (instead of just one or two datasets) are considered for each training dataset to better illustrate the advantage of DIMAML.

In addition, the training on the test tasks are conducted for a maximum of 100 epochs (some tasks are less), which seems a little bit not enough. I'm not sure about how many epochs are required for convergence on each test task, and suggest the authors to state this info more clearly in the experiment section, and run all the test tasks to convergence if not yet, thus we can have a fairer comparison with other methods w.r.t. the model's final performance.

P.S. a typo: Figure.2 should be Table.2

---

### Official Review · Reviewer_opzS · 2021-06-17
**Review for Discovering Weight Initializers with Meta Learning**

**Rating:** 7
**Confidence:** 3

**Review:**

This paper is to improve neural network weight initialisation from a meta-learning perspective. The author proposes a novel meta-learning approach to learn the weight initialisation for a given architecture by sampling from parametric and differentiable distribution. First, they assume the initial weight to be i.i.d and sampled from the distribution with trainable parameters. Then, the network will be trained with multiple steps in the inner loop, and the weight initializer will be updated via minimising the average validation loss. Regarding the choice of the differentiable distribution, the author proposes to use either normal distribution or the parameterized quantile function. The normal distribution could be differentiable, but it limits the space of the meta-algorithm to learn. The latter is to learn the shape of weight distribution and use the piecewise linear increasing function to approximate the continuous quantile function with the slope regions. They evaluate their result on both computer vision and language dataset. They further show that their algorithm has comparable performance on the state-of-the-art meta learned initializer.

Pros:
- This is novel approach to learn the weight initialization from meta-learning perspective
- Compared to the previous MetaInit, the proposed objective function of average validation loss is more intuitive and supposed to lead to faster learning and guarantee optimal training performance.
- They also show comparable performance against the state-of-the-art MetaInit.

Cons:
- A detailed description of the inner loop and outer loop of DIMAML is missing. I recommend the author provide more description of how the inner loop, outer loop, and the entire training procedure work in the paper or appendix.
- The detail of the transferability experiment is missing. It will be interesting to see more experiments on some other unseen datasets.

---

### Decision · Program_Chairs · 2021-06-21

Accept (Oral)